# Autoimmune Disorders & COVID-19

**DOI:** 10.3390/medicines8100055

**Published:** 2021-09-28

**Authors:** Leonardo Freire-de-Lima, Aline Miranda Scovino, Camilla Cristie Barreto Menezes, Leonardo Marques da Fonseca, Jhenifer Santos dos Reis, Marcos André Rodrigues da Costa Santos, Kelli Monteiro da Costa, Carlos Antonio do Nascimento Santos, Celio Geraldo Freire-de-Lima, Alexandre Morrot

**Affiliations:** 1Instituto de Biofisica Carlos Chagas Filho, Universidade Federal do Rio de Janeiro, Rio de Janeiro 21941-170, Brazil; lfonseca@biof.ufrj.br (L.M.d.F.); jhnffrrs8@gmail.com (J.S.d.R.); rodrigues8mr@gmail.com (M.A.R.d.C.S.); kellimc85@gmail.com (K.M.d.C.); cansantos.bio@gmail.com (C.A.d.N.S.); celio@biof.ufrj.br (C.G.F.-d.-L.); 2Instituto de Microbiologia Paulo de Góes, Universidade Federal do Rio de Janeiro, Rio de Janeiro 21941-170, Brazil; ali.scvino@gmail.com; 3Instituto Oswaldo Cruz, FIOCRUZ, Rio de Janeiro 21040-360, Brazil; ccristie.95@gmail.com; 4Faculdade de Medicina, Universidade Federal do Rio de Janeiro, Rio de Janeiro 21044-020, Brazil

**Keywords:** coronavirus disease 19, autoimmunity, viral infections, immune system

## Abstract

Coronavirus disease 2019 (COVID-19) can progress to severe pneumonia with respiratory failure and is aggravated by the deregulation of the immune system causing an excessive inflammation including the cytokine storm. Since 2019, several studies regarding the interplay between autoimmune diseases and COVID-19 infections is increasing all over the world. In addition, thanks to new scientific findings, we actually know better why certain conditions are considered a higher risk in both situations. There are instances when having an autoimmune disease increases susceptibility to COVID-19 complications, such as when autoantibodies capable of neutralizing type I IFN are present, and other situations in which having COVID-19 infection precedes the appearance of various autoimmune and autoinflammatory diseases, including multisystem inflammatory syndrome in children (MIS-C), Guillain-Barré syndrome, and Autoimmune haemolytic anaemia (AIHA), thus, adding to the growing mystery surrounding the SARS-CoV-2 virus and raising questions about the nature of its link with autoimmune and autoinflammatory sequelae. Herein, we discuss the role of host and virus genetics and some possible immunological mechanisms that might lead to the disease aggravation.

The distinction between what is self or non-self by the immune system is not absolute, and in some situations, it can make mistakes, leading to inflammatory diseases called autoimmune diseases. In a very simplified way, autoimmunity can be defined as a reaction of the immune system to self-antigens. The autoreactivity itself can be physiological, necessary for the positive selection of lymphocytes and efficient response against infections, tumors and autoimmune diseases themselves, or pathogenic, leading to tissue and organ damage [1]. Most autoimmune diseases are multifactorial, determined by both polygenic and environmental factors [2]. The influence of the microbiome in the predisposition to these diseases is also a factor worth considering [3]. Generally, genetic susceptibility results from the additive effects of several common risk variants, each with small effects that alone are insufficient. More than 100 gene loci related to rheumatoid arthritis, multiple sclerosis, and inflammatory bowel diseases have been described. Many of them overlap, suggesting the participation of common immune mechanisms in the development of these diseases, such as the PTPN22, CTLA4, IL23R, and TYK2 genes [4]. It is well established that some infections can trigger the development of autoimmune diseases, and some mechanisms are proposed to explain this phenomenon. One of them is the mimicry of host antigens by pathogens. The molecular similarity between the pathogen and host antigens alone is not sufficient for the development of autoimmunity. It will occur when the host has molecules of the MHC complex capable of binding and presenting the previously processed antigen. In addition, it is necessary to have peripheral self-reactive B or T cells for the antigen, which must escape negative selection. Those factors must also lead to an immune response intense enough to produce high levels of cytokines after the recognition of the viral antigen [5]. It is well described in the literature that infection with the hepatitis B virus is associated with the development of multiple sclerosis. Hepatitis B virus has some molecules that mimic self-molecules, such as the viral DNA polymerase that resembles human nuclear and muscle proteins [6]. Another example is the infection by group A beta-hemolytic streptococci, that is associated with rheumatic fever. In this case, there is mimetism between the bacterial M protein and cardiac myosin, leading the immune system to target both, and causing inflammation in the heart. [5]. Other mechanisms are epitope spreading and presentation of cryptic antigens. Epitope spreading is defined as the diversification of epitopes recognized by T and B lymphocytes. Experiments conducted in mice identified the induction of Systemic Lupus Erythematosus after immunization with a peptide from the EBNA-1 protein of the Epstein-Barr virus [7]. On the other hand, the presentation of cryptic antigens can happen when during an infection by a microorganism, these antigens are processed and presented. Viral infections can cause tissue damage that exposes previously hidden self-antigens, the cryptic antigens. Eventually, these antigens can become the target of an immune response. The elicited inflammatory response intensifies the presentation of these antigens, which are then presented to autoreactive T lymphocytes, initiating the autoimmune response [8]. Hepatitis B virus infection, in addition to being able to induce autoimmunity by molecular mimicry, can also expose intracellular antigens, which would not be exposed in the absence of the infection [6]. As we can see, the association of viral infections and the development of autoimmunity is well established in the literature. The same has been observed with the new SARS-CoV-2 virus, since the beginning of the pandemic. This virus appeared in late 2019 in China, and quickly spread throughout the world, being today, according to the World Health Organization, responsible for the death of almost five million people worldwide [9]. Although some risk factors are already associated with a greater likelihood of worsening of the disease, such as age, gender, cardiovascular disease and other comorbidities, we still do not fully understand why some patients develop the severe form of the disease, while others are asymptomatic [10,11]. The high mortality rate seen in COVID-19 is related to the unregulated activation of the immune system. It is known that the immune response is crucial to resolve coronavirus infections and also that unbalanced immune reactions may lead to immunopathologies and impaired pulmonary gas exchange [12]. Patients who evolve to the severe form of the infection have a high neutrophil/lymphocyte rate, acute pulmonary neutrophilic infiltration showing elevated serum cytokines, ferritin, hemophagocytosis, D-dimer, and soluble CD25 (the interleukin [IL]-2 receptor alpha chain). The presence of activated neutrophils and macrophages in the target tissues has been associated with induction of neutrophil extracellular traps (NETs) and thrombocytogenesis, promoting vascular collapse, respiratory distress, and multiorgan failure, which are related to the so-called cytokine release syndrome (“cytokine storm”), including excessive productions of granulocyte and macrophage colony-stimulating factor (GM-CSF), IL-2, IL-6, IL-7, IL-10, tumor necrosis factor α (TNF-α), and granulocyte colony-stimulating factor (G-CSF) [13]. We know that children respond better to viral infections, due to a trained immune system (both due to vaccines and recurrent infections). This is no different for SARS-CoV-2, with children typically having a better prognosis, which may also be due to the absence of comorbidities, cross-reactive immunity to common cold coronaviruses, and the reduced number of ACE receptors in their noses. In addition, patients with chronic comorbidities, which are more common in the elderly population, have an increased pro-inflammatory baseline environment, which make them more susceptible to immune dysregulation. Moreover, aged patients have some differences in the immune response, which release the increased risk of showing a cytokine storm [14]. In the more severely ill patients, there was elevation of inflammatory cytokines suggestive of immune dysregulation and possible autoimmunity. Although type I IFNs are traditionally effective against viral infections, including against SARS-CoV-2, some studies show that exacerbated or inefficient IFN response may be the cause of more severe cases of the disease. It has been reported that the viral ORF6, ORF8, and nucleocapsid proteins play an important role in modulating the host innate immunity. They are potential inhibitors of type I IFN-β and NF-κβ-responsive promoter, an innate immune signaling pathway critical for the host defense against viral infections. Low levels of type I interferons probably lead the immune system to compensate with unregulated activation of responses in the acute phase of infection, as exemplified by cytokine storm. In general, the predisposing factors for development of the cytokine storm consist of a diverse combination of mechanisms, involving viral escape associated with genetic defects of host defense, as well as other immunological abnormalities, such as high rate of neutrophil infiltration into target tissues [13]. Zhang et al. identified deleterious mutations in IFN genes that impede their function in patients with the severe form of the disease. The viral loads were higher in these patients than healthy donors, demonstrating an inability to properly clear the virus. Together, these data convey the importance of type I IFN signaling in defense against SARS-CoV-2 infection and suggest that inherited deleterious variants explain a subset of severe COVID-19 cases [15]. In a recent study with 987 patients hospitalized for the treatment of pneumonia caused by SARS-CoV-2, they observed that at least 101 patients had autoantibodies against type I interferons (IFN-I), most being male. These autoantibodies are able to neutralize IFN-I, preventing its effects and leading to more severe forms of the disease. In addition, 40 patients who had autoantibodies against all 13 types of IFN-α, had undetectable levels of them in the plasma, indicating the neutralizing action of these autoantibodies also in vivo. At the same time, these autoantibodies were not found in the 663 asymptomatic patients or those who developed the mild and moderate forms of the disease, and in only four of the 1227 healthy individuals analyzed in this study [16]. The data indicate that the presence of autoantibodies precedes infection, and would be a risk factor for the development of severe forms of the disease. In the same study, they were able to assess the presence of these autoantibodies in two patients before infection, and both patients developed the severe form of the disease. Likewise, they evaluated three patients with autoimmune polyendocrinopathy syndrome type I (APS-1), an autoimmune syndrome that naturally has the production of autoantibodies against IFN-I. These patients also developed severe forms of COVID-19 [16]. Other works support this hypothesis, as the first work published in 1984 showing that the presence of autoantibodies against IFN made people more susceptible to varicella-zooster [17]. In addition, more recent work has shown the presence of autoantibodies against various immune target antigens in patients with severe forms of COVID-19, including against IFN-I [18]. The presence of autoantibodies in critically ill COVID-19 patients is not restricted to those associated with IFN-I. Studies have linked the infection by the SARS-CoV-2 virus with the presence of autoantibodies associated with antigens present in certain tissues that may explain the development of autoimmune and inflammatory multisystemic diseases. Among the antigens described are some associated with the central nervous system, the vascular endothelium and the extracellular matrix [18]. Some studies describe the association of Guillain-Barré Syndrome with COVID-19. In general, they describe the onset of neurological symptoms after 7 to 10 days of the respiratory symptoms characteristic of COVID-19. However, it is not yet known whether infection by SARS-CoV-2 induces the production of autoantibodies against ganglisiosides in the peripheral nervous system, as occurs with other infections [19,20,21]. Vascular events associated with COVID-19, such as venous and arterial thrombosis or stroke related to the presence of autoantibodies against phospholipids have been described and observed in some patients [22]. High levels of these antibodies have been associated with increased neutrophils and NETs release, increased platelet levels, greater respiratory complications, and kidney problems [23]. Analysis of the serum of patients who died of COVID-19 shows elevated levels of autoantibodies against Annexin A2, an important protein for the maintenance of the microvasculature of the lungs [24]. Another disease that has been associated with COVID-19 is the Multisystem Inflammatory Syndrome in children (MIS-C). This syndrome is similar to Kawasaki’s disease (KD), which is a systemic vasculitis that usually affects children under five years old and although self-limited, it can cause coronary artery aneurysm in a high percentage of cases [25]. Despite some very similar symptoms, MIS-C has some peculiar characteristics. First, the children affected by MIS-C are significantly older than those diagnosed with KD. Other characteristics such as leukopenia, high levels of ferritin, C-reactive protein (CRP) and ventricular natriuretic peptide (a marker of heart failure), as well as low platelet count are not characteristic of KD [26]. Since the beginning of the pandemic, they have observed an unusually high number of critically ill patients with clinical features consistent with KD. The occurrence of such cases with a stereotyped clinical picture, which is rarely seen in children, during the peak of SARS-CoV2 infection in highly epidemic areas, points to a causal relationship. In fact, it is already known that some infections can trigger the disease such as the Epstein Barr virus, Cytomegalovirus, other Coronaviruses, staphylococci, streptococci, etc. The delay between the pandemic peak of COVID-19 and the occurrence of this hyperinflammatory syndrome in children does not point to a direct effect of the infection, but rather to an immune-mediated disease [27]. MIS-C is a hyperinflammatory condition associated with COVID-19. The pathological mechanisms of SARS-CoV-2 infection responsible for MIS-C are not known for sure. Some authors propose that it is not associated with the infection itself, but with later phenomena, such as the production of antibodies that induce antibody-dependent enhancement (ADE), or the blocking of IFNs I and III by the virus can cause a late cytokine storm, which will be the cause of MIS-C [26]. Some others possible mechanisms by which SARS-CoV-2 could induce this hyperinflammatory syndrome would be the formation of autoantibodies by molecular mimicry and vascular damage secondary to the deposition of immune complexes as in acute serum disease. Today, some clinical trials show good results when using drugs classically used for the treatment of rheumatic diseases, such as intravenous corticosteroids and intravenous immunoglobulins, both in the treatment of severe cases of COVID-19 and for the treatment of MIS-C, in the attempt of try managing immunologic complications caused by the infection [27]. Although SARS-CoV-2 infection appears to occur asymptomatically in most children, there is growing evidence that it can cause a systemic hyperinflammatory syndrome that mimics KD but has some peculiar characteristics. The role of host and virus genetics and the exact immunological mechanisms that lead to the disease are far from being understood [28]. We can see that autoimmune disorders have two important and distinct roles in COVID-19, on the one hand the presence of autoantibodies, such as those against IFNs [15], have a predictive role in relation to disease severity and may be related to severe cases of the disease. On the other hand, there is some evidence that the infection itself triggers autoimmunity, such as those seen in relation to Guillain-Barré Syndrome [18,19,20], to autoantibodies against Annexin A2 [23], and as observed in some studies, the positive association between the infection by SARS-CoV-2 and thrombocytopenic purpura [29], autoimmune haemolytic anaemia [30] and Miller Fisher syndrome (MFS) (a mild variant of Guillain-Barré syndrome) [27]. Thus, we can say that COVID-19 is a complex disease, not only in its cause, but also with serious and significant chronic consequences, which go beyond those closely related to infection, such as lung, renal, or cardiovascular damage. More studies are needed to understand the causal relationships between infections and autoimmune disorders, in order to mitigate these consequences.

## Data Availability

Not applicable.

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
