# Peer review of "Autoimmune Disorders & COVID-19"

_medicines, 2021, doi:10.3390/medicines8100055_

Round 1

Reviewer 1 Report

The manuscript is very interesting, authors give their comments on the possible relationship between COVID-19 and autoimmune diseases, based on previous scientific knowledge and experiences and hypothesize about the possible mechanisms by which SARS-Cov-2 could induce the hyperinflammatory syndrome, which suggest where further studies should be directed in this concern. The manuscript is written using a clear and concise language.

Author Response

Thank you very much for your positive comments. The English language has been reviewed by a native speaker as suggested by the reviewer.

Reviewer 2 Report

To the Authors,

I have reviewed with interest this paper.

The paper is of potential interest, but has some limitations.

Firstly, I recommend the help of a native English speaker with experience in scientific writing, as sometimes the paper is hard to follow and some sentences are not clear.

Major comments:

1) This work analyses different aspects related to the immune response to SARS-CoV-2 and the association between the infection and autoimmunity. However, as presented, it does not add a new point of view to current knowledge. I suggest to better stress the main point with relevance for this paper:

-The main features of the immune response against coronaviruses, with a particular focus on SARS-CoV-2 (please see doi: 10.1002/jmv.25685). To this regard, the relevance of IFN should be stressed (see also Zhang et al., Science 370, eabd4570 (2020))

-The mechanisms underlying the development of the immune dysregulation in patients with severe COVID-19 and MIS-C

-The mechanisms responsible for the potential association between COVID-19 and autoimmunity

-A deeper conclusion. Are there new clinical and therapeutic implications?

 2) The mechanisms responsible for the development of autoimmunity, with specific focus on the role of infectious triggers, should be better discussed

3) L 60: other studies analysed the role of age and comorbidities, with specific focus on their influence on the immune response. See: doi: 10.1002/iid3.404.

It would be interesting to add that patients with chronic comorbidities have an increased pro-inflammatory baseline environment, which make them more susceptible to immune dysregulation. Moreover, aged patients have some relevant differences in the immune response, which are relevant for the increased risk of showing a cytokine "storm" 

4) some important elements to be specified on MIS-C (that has to be correctly spelled and abbreviated):

-it can present with KD-like features, but it is not mandatory (doi: 10.1016/S2352-4642(20)30304-7)

-Please also see current diagnostic criteria for MIS-C.

-MIS-C can develop in a different age-range (higher) compared to classic KD

-the pathogenesis of MIS-C is not clearly elucidated, but altered IFN response seems to have a pivotal role (see doi: 10.1016/j.cell.2020.09.016 ; doi: 10.1038/s41577-020-0367-5)

5) another aspect to point out, although not the main focus of this paper, is that drugs used in autoimmune disorders (anti-cytokine agents, and others) have demonstrated efficacy in severe COVID-19 and MISC-C (see DOI: 10.1007/s10067-020-05190-5;  doi: 10.1186/s12969-021-00559-5; doi: 10.1016/j.autrev.2020.102523. ). 

6) Is the word “opinion” really necessary in the title?

7) The clinical relevance of this opinion paper can be better presented. Is there a higher susceptibility of severe COVID-19 in patients with autoimmune disorders (i-e arthritis, SLE)/being treated with immunosuppressive agents?Are there clinical data regarding the incidence of autoimmune disorders presenting after/with COVID-19?

8) The abstract should be improved and should better represent the content of the manuscript

Minor comments:

  • L 49 please use "in mice"
  • L 59 please specify the date in which data were lastly updated by WHO
  • Please uniform the use of "COVID-19" through the text
  • L 53: what does "current inflammatory response" mean?
  • L 98: please specify the entire name term "NETS"

I hope that the authors will find this comments as helpful and constructive

Author Response

Dear reviewer, we appreciate your positive comments. As suggested, the text has been reviewed by a native English speaker.

Regarding the major comments were addressed according to the referee's suggestions:

1) Please, see lines 80-97; lines 102-120.

2) Please, see lines 54-56; 58-59; 65-67 and 69-71.

3) Please, see lines 97-102.

4) Please, see lines 157-158; lines 161-165.

5) Please, see lines 182-186

6) The word "Opinion" has been removed from the title.

7) Please, see lines 190-202.

8) The abstract has been improved to attend the referee's suggestion.

Minor comments:

  • The sentence has been rewritten (now lines 61-63) 
  • Regarding the data about COVID-19 deaths, the WHO website is updated every week, so we included the most recent figure, from September, 20. The link to the data in the website, as well as the date we accessed the information are available in the reference section (now lines 74-77)
  • The expression "COVID-19" was used in that form throughout the text.
  • The expression"current inflammatory response" has been changed to The "elicited inflammatory response"
  • The meaning of the acronym was specified in line 88.

We would like to that the reviewer for their time and suggestions to improve the quality of the manuscript.

Round 2

Reviewer 2 Report

The authors have addressed most of the comments. The paper can be accepted for publication